# Dementia in UK South Asians: a scoping review of the literature

Amy Blakemore,[1] Cassandra Kenning,[2] Nadine Mirza,[2] Gavin Daker-White,[3] Maria Panagioti,[2] Waquas Waheed[2]

[1]Division of Nursing, Social Work and Midwifery, School of Health Sciences, Faculty of Biology, Medicine and Health, Manchester Academic Health Science Centre, The University of Manchester, Manchester, UK
[2]Division of Population Health, Health Services Research and Primary Care, School of Health Sciences, Manchester Academic Health Science Centre, The University of Manchester, Manchester, UK
[3]NIHR Greater Manchester Primary Care Patient Safety Translational Research Centre, Manchester Academic Health Science Centre, The University of Manchester, Manchester, UK

**Correspondence to**
Dr Amy Blakemore; amy.blakemore@manchester.ac.uk

## ABSTRACT

**Objective** Over 850 000 people live with dementia in the UK. A proportion of these people are South Asians, who make up over 5% of the total UK population. Little is known about the prevalence, experience and treatment of dementia in the UK South Asian population. The aim of this scoping review is to identify dementia studies conducted in the UK South Asian population to highlight gaps in the literature which need to be addressed in future research.

**Method** Databases were systematically searched using a comprehensive search strategy to identify studies. A methodological framework for conducting scoping reviews was followed. An extraction form was developed to chart data and collate study characteristics and findings. Studies were then grouped into six categories: prevalence and characteristics; diagnosis validation and screening; knowledge, understanding and attitudes; help-seeking; experience of dementia; service organisation and delivery.

**Results** A total of 6483 studies were identified, 27 studies were eligible for inclusion in the scoping review. We found that studies of prevalence, diagnosis and service organisation and delivery in UK South Asians are limited. We did not find any clinical trials of culturally appropriate interventions for South Asians with dementia in the UK. The existing evidence comes from small-scale service evaluations and case studies.

**Conclusions** This is the first scoping review of the literature to identify priority areas for research to improve care for UK South Asians with dementia. Future research should first focus on developing and validating culturally appropriate diagnostic tools for the UK South Asians and then conducting high-quality epidemiological studies in order to accurately identify the prevalence of dementia in this group. The cultural adaptation of interventions for dementia and testing in randomised controlled trials is also vital to ensure that there are appropriate treatments available for the UK South Asians to access.

## INTRODUCTION

As the global population ages the prevalence of dementia increases; there are an estimated 47 million people living with dementia worldwide and this figure is set to increase to over 75.6 million by 2030.[1 2] In the UK alone, there are over 850 000 people living with dementia at a cost of £26 billion per year.[1] The rising prevalence of dementia and its associated cost and burden has generated an increased

focus on the timely screening, diagnosis and treatment of dementia.

Ethnic minority groups make up over 14% of the total UK population, many of whom are South Asian (Pakistan, India and Bangladesh). In the 2001 census, South Asians were 3.9% of the total UK population and by 2011 this figure had risen to 5.3%.[3] It is estimated that there are over 25 000 people from ethnic minority groups living with dementia in the UK.[4] However, the true prevalence of dementia within the UK South Asian community is yet to be established. As the overall prevalence of dementia in the UK increases so too will the prevalence among the growing population of UK South Asians. Increased prevalence places an increased burden on healthcare services to understand the presentation of dementia in UK South Asians in order to identify, diagnose, and treat this population.

There are difficulties in identifying, diagnosing, and treating dementia in UK ethnic minority groups, including the South Asian population.[5 6] This is due to low levels of literacy, language barriers and a lack of appropriately translated and culturally adapted screening and diagnostic tools for this ethnic group.[7] For example, in the South

Asian community only 35% of older people (aged over 65 years) can speak English and only 21% can read and write English, with most relying instead on their first language, which for many is Urdu.[3 8 9] This makes the completion of screening tools and diagnostic tests challenging. In addition to barriers of language and literacy, there is a lack of awareness about dementia within the South Asian community. Many South Asian people view memory loss as a normal part of ageing or understand symptoms of dementia by religious belief.[7] This difference in the understanding and explanation of the presentation of dementia may result in reduced help-seeking.[7] Lack of help-seeking prohibits early intervention and treatment, which is important to reduce the burden of dementia for patients, family carers, and the wider healthcare system.[5 6 10] If we are to encourage UK South Asians to seek help early and improve our ability to engage this group with formal healthcare for dementia, we need to provide culturally appropriate services for them to access. Currently, there are a lack of both culturally appropriate services and accurately translated neuropsychological assessments for UK South Asians.[11]

In order to identify the scope of the literature on dementia in the UK South Asian population, we have conducted a scoping review. Scoping reviews are designed to assess the nature and extent of the literature available on a particular topic.[12] This scoping review aims to identify where there are gaps in the evidence around the prevalence, identification, diagnosis and treatment of dementia in the UK South Asian population.[13] This will then allow recommendations to be made for future research; the results of which we hope will inform the identification, diagnosis, and treatment of dementia in UK South Asians.

## METHODS

To examine the literature on dementia in UK South Asians, we conducted a scoping review in accordance with the methodological framework for scoping reviews published by Arskey and O'Malley and further developed by Levac *et al.*[14 15] The Arksey and O'Malley framework was found to be the most commonly used methodology for scoping reviews in a systematic review of scoping methods.[13] The framework includes guidance on the following areas, which are outlined in the sections below: identifying the research question, searching for relevant studies, selecting studies, charting the data and collating and summarising results.

### Inclusion criteria and exclusion criteria

Studies that had included UK South Asian patients with dementia were eligible for inclusion, as well as those that had focused on family carers, and healthcare professionals working with patients with dementia. We did not exclude studies based on year of publication. Published dissertations were included but any unpublished dissertation was not. Systematic reviews and narrative literature reviews were not included but we hand-searched the reference lists of all the relevant reviews, which were returned in the search in order to identity additional primary studies for inclusion. Conference proceedings were not included.

We excluded studies that did not report on South Asian participants alone, as a comparator group or where data could not be separated from data for participants of other ethnicities. We excluded studies that were not published in the English language.

### Search strategy

Search terms were kept broad to identify the maximum number of studies that were eligible for inclusion. The search strategy was developed within the team to identify all studies relevant to ethnicity and dementia, this was then refined through hand checking to those studies which included UK South Asian participants. Search terms included: Dementia, Alzheimer* Disease; ethnic*, Asian, Black, African, minority, ethnic group, multi-ethnic. See online supplementary material 1 for the full search strategy.

The search was conducted without a study design filter in order to retrieve studies using all methodologies, including: systematic reviews, qualitative studies, quantitative studies, case studies, and service evaluations.

The following databases were searched: Cochrane Register of Controlled Trials (CENTRAL), MEDLINE (Ovid), PsycINFO (Ovid), Embase (Ovid), Cochrane database of systematic Reviews. The search was initially conducted in April 2016 and updated in June 2017. In addition to the database searches, we hand-searched the reference lists of all relevant systematic reviews and literature reviews that were returned in the search to identify any additional papers for inclusion.

### Screening and charting the data

Electronic search results were managed using EndNote X7 and Microsoft Excel. Titles and abstracts of all citations were first screened by author (AB), those that were not related to dementia and South Asians, or had been conducted outside of the UK, were discarded. The full text of all potentially eligible papers was then obtained and assessed against the inclusion criteria (AB and WW). Where the full text of papers was not available we contacted authors to request the paper. Any ambiguities about whether or not a study met the inclusion criteria were resolved by discussion at a meeting attended by all authors.

Studies that met the review criteria were charted using a data extraction sheet designed by AB and WW. Data were primarily extracted by AB and reviewed by WW. Data were extracted for the following domains: year of publication, study design, ethnicity, aim of study, study setting (community/primary care/hospital), eligibility criteria, participants (patient/carer/healthcare staff), type of dementia, age of participants, sample size, main findings.

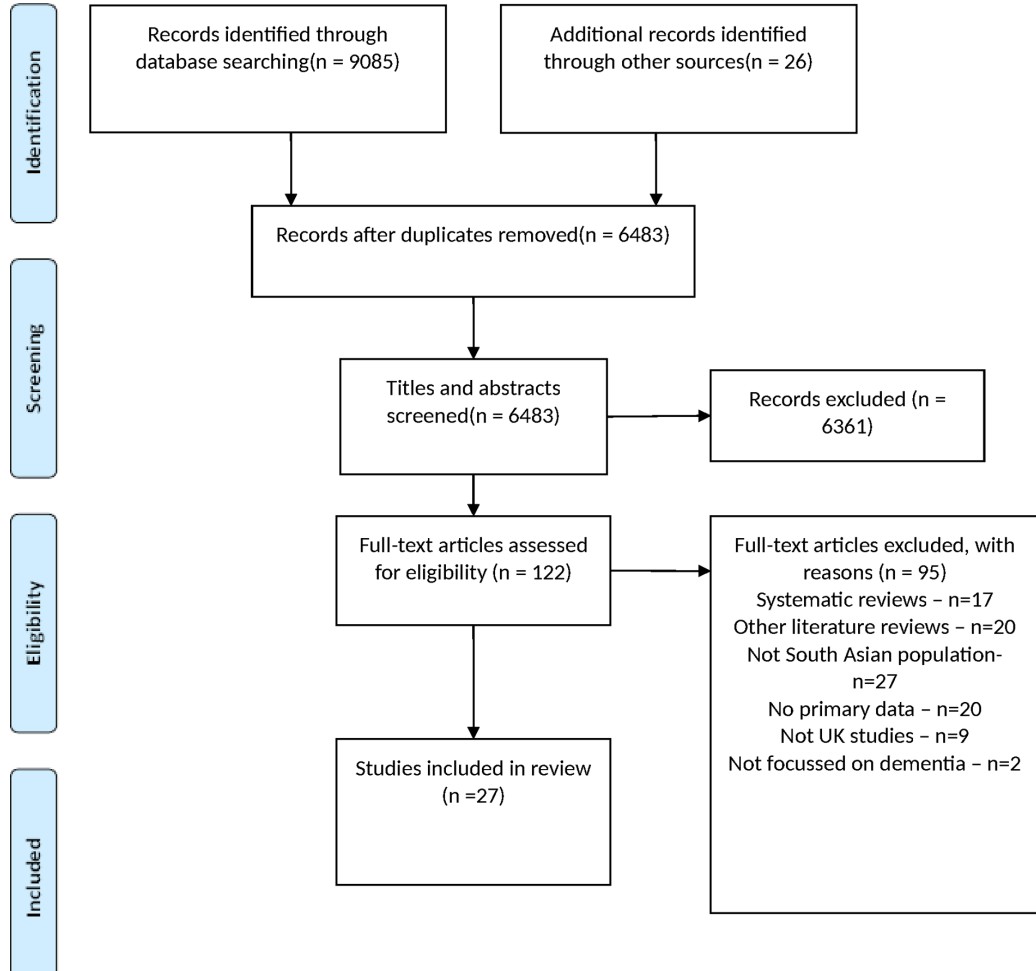

**Figure 1** Flow of included studies.

## Data synthesis

Once the pool of eligible studies was agreed, they were grouped into categories according to their primary focus (AB, CK, WW). These categories were defined as: prevalence and characteristics; service organisation and delivery; views and experiences; validation, screening and diagnosis. All data are reported in a narrative format. As this is a scoping review, there is no formal review of the quality of the included literature.[14 15]

## RESULTS
### Description of included studies

The initial search identified 6483 studies after duplicates were removed. Screening of the titles and abstracts left 122 studies which met the criteria for full-text review, 26 were then selected for inclusion.[16–41] To update the review, the search was rerun in June 2017; one new paper was found to be eligible for inclusion, bringing the total number of included studies to 27.[42] Figure 1 shows the flow of study selection and numbers of excluded studies with reasons.[43]

Table 1 shows a summary of the findings for all 27 included studies. For the purpose of narrative review, we grouped the studies into six categories: prevalence and characteristics; diagnosis validation and screening; knowledge, understanding and attitudes; help-seeking; experience of dementia; service organisation and delivery.

### Prevalence and characteristics

We identified five articles that had looked at either the prevalence or characteristics of dementia in the UK South Asians.[18 21 31 39 41]

Bhatnagar and Frank looked at the prevalence of dementia in a community sample of people from the Indian subcontinent living in Bradford, UK.[18] Diagnostic interviews were conducted with 100 participants and the prevalence of dementia was reported as 4%. However, when tested using a Hindi translation of the Geriatric Mental State (GMS-A), which found an initial prevalence of 7%, the reliability of the diagnosis was poor. The authors suggest that this is due to a lack of cultural adaptation of the measure. For example, some interviewees would have little understanding of western calendar months, a knowledge of which forms part of the screening measure.[18]

A cross-sectional study conducted in inner-city Liverpool also used the GMS interview to assess the prevalence of dementia in English-speaking versus non-English-speaking

**Table 1** Summary of characteristics and findings of included studies

| Author | Year | Study design | Total sample size | Sample | Setting | Summary of main findings |
|---|---|---|---|---|---|---|
| Adamson[16] | 2001 | Qualitative | 30 | n=12 (40%) South Asian carers of people with dementia | Healthcare services (carer support, day centres, psychiatric services) | Lack of knowledge of dementia found in South Asian group. South Asian participants talked about symptoms being a result of past actions in life and apportioning blame. They also believed that other physical conditions and their associated medications could cause dementia, such as antidepressants for depression. |
| Adamson and Donovan[17] | 2005 | Qualitative | 36 | n=15 (42%) South Asian carers of people with dementia | Community | South Asian participants talked about caring for family as a cultural norm and wider families were more likely to live together to facilitate this. |
| Bhatnagar and Frank[18] | 1997 | Cross-sectional study | 100 | Aged 65–89 years, from Indian subcontinent and living in Bradford | Community | Prevalence of dementia 4% as diagnosed by psychiatrist and 7% using Hindi translation of diagnostic measure (GMS-A). |
| Bowes and Wilkinson[19] | 2003 | Qualitative | 11 | 11 interviews with carers 4 case studies of South Asian patients | Community | Carer interviews: demand for services, a need to develop awareness and knowledge in the community and to promote a culturally sensitive response from services. Case studies: negative experiences of dementia, poor quality of life, need for support, lack of access to appropriate services, little knowledge about dementia, isolation from both community and family life. |
| Giebel et al[20] | 2016 | Mixed method pilot | 33 | Three groups—South Asian, over 60 years: without memory problems; memory problems not consulted GP; memory problems had consulted GP | Community | Those who had not consulted a GP often considered memory problems to be given by God and did not identify medical support as appropriate for them. Those who had attended a consultation with GP identified forgetfulness and loss of social meaning as symptoms of dementia. |
| Giebel et al[42] | 2016 | Questionnaire validation | 25 | n=25 South Asian | Community | 123 new perceptions around South Asian their understanding of dementia were identified. These were added to the BEMI-C to create a new checklist (BEMI-D). |
| Haider and Shah[21] | 2004 | Pilot study | 62 | n=31 South Asian, aged 65–96 years n=31 white British, aged 65–90 years | Day hospital | South Asian participants score lower on the BEHAVE-AD phobia and anxiety subscale. Alzheimer's disease associated with vascular dementia with affective disturbance. |
| Hailstone et al[22] | 2016 | Questionnaire validation | 58 | Mean age 60 years 59% female (n=34) First-generation and second-generation South Asians | Community | Strongest predictor of willingness to seek help for dementia was perceived social pressures from significant others. Attitudes in the questionnaire predicted 77% of variance in willingness to seek help, but no relationship was found with dementia knowledge. |
| Jutla[23] | 2015 | Qualitative | 12 | South Asian Sikhs caring for someone with dementia and living in Wolverhampton, UK | Community | Understandings participant's migration experiences and identities is important for understanding family carers experience of services when caring for someone with dementia. |
| Kaur et al[24] | 2010 | Service evaluation | NA | An Asian link nurse working in Wolverhampton, UK | Community mental health team | Having an Asian link nurse was vital in providing education about dementia for South Asian people. |

Continued

**Table 1** Continued

| Author | Year | Study design | Total sample size | Sample | Setting | Summary of main findings |
|---|---|---|---|---|---|---|
| La Fontaine et al[25] | 2007 | Qualitative | 49 | South Asians aged 17–60 years who were English, Hindi or Punjabi speaking | Community | Interviews highlighted that cognitive impairment was rarely mentioned when talking about ageing. Ageing was seen as a time of withdrawal and isolation. There was a sense of stigma and a lack of knowledge about mental health services, which leads to exclusion from these services. |
| Lawrence et al[26] | 2008 | Qualitative | 32 | n=10 (31%) South Asian carers of people with dementia | Community | South Asian carers possessed a traditional caregiver ideology, conceptualising caregiving as natural, expected and virtuous. This informed their attitudes towards formal healthcare services. |
| Lawrence et al[27] | 2011 | Qualitative | 30 | n=9 (30%) South Asian Aged 67–87 years | Mental health services | Interviews highlighted that participants engaged in a process of appraisal where they assessed how much their condition affected valued elements of their life. |
| Lindesay et al[28] | 1997 | Questionnaire validation | 1297 | n=149 (11%) South Asian, Gujarati | General practice | Mean MMSE scores were lower in the Gujarati group due to the effects of age, education and visial impairment. The MMSE performed comparably in both groups as a screen for moderate-to-severe dementia but was less effective for Gujaratis with mild dementia. |
| Mackenzie[29] | 2006 | Qualitative | 21 | n=16 (76%) South Asian carers of people with dementia | Community | In the South Asian group stigma was linked to religious and magical explanations for the onset of dementia, which affected the ability of carers to access support. |
| McCracken et al[41] | 1997 | Cross-sectional study | 579 | n=12 (2%) Asian aged over 65 years | Community | Prevalence of dementia 9% among English-speaking Asian participants. |
| Mukadam et al[30] | 2015 | Qualitative | 53 | South Asians aged 18–83 years | Community | Stigma around dementia was linked to ideas of 'madness' a lack of physical explanations and a lack of treatment. Barriers to help-seeking were that memory problems were an inevitable part of ageing. Denial of symptoms was evident in order to maintain position in the family and community, and due to fear of institutionalisation. |
| Odutoye and Shah[31] | 1999 | Cross-sectional study | 242 | n=29 (12%) South Asians newly referred to psychogeriatric unit between 1995 and 1997 aged 58–96 years | Psychogeriatric unit | South Asians were less likely to have dementia than white British elders ($X^2$=5.05, 1 df, P<0.03). |
| Purandare et al[32] | 2007 | Cross-sectional study | 246 | n=191 (78%) South Asian, mean age 72.4 years (SD 10.6) | Community–day centre | Knowledge of dementia was poor in both South Asian and white British people. South Asians had less knowledge about basic aspects of dementia (P<0.001) and the epidemiology of dementia (P<0.001) as compared with white British people. |
| Rait et al[33] | 2000 | Validation of screening instrument | 120 | Community resident South Asians aged over 60 years. n=65 Gujarati speaking, mean age 70 years (SD 6.8) n=39 Pakistani group, mean age 68 years (SD 6.0) | Community | Both modified screening tests (MMSE and AMT) had high sensitivity scores but ethnic background was found to influence the cut-off scores for these measures. The MMSE cut-off score was found to be significantly higher in the Pakistani group (≥27, sensitivity 100%, specificity 95%) compared with the Gujarati group (≥24, sensitivity 100%, specificity 77%). |

Continued

**Table 1** Continued

| Author | Year | Study design | Total sample size | Sample | Setting | Summary of main findings |
|---|---|---|---|---|---|---|
| Regan[40] | 2016 | Case study | NA | Case study of a male Muslim patient with young onset frontotemporal dementia | Dementia services | Mostly negative experiences of accessing services and an inability to access support from either family or the religious community. Services not equipped to support people with young onset dementia from an ethnic minority. |
| Seabrooke and Milne[34] | 2009 | Service pilot | 4 | South Asian patients aged 65–93 years with memory problems | Primary care | Inviting older Asian patients with memory problems to see a specially trained Asian nurse using a culturally appropriate information leaflet encouraged a small number of people to access the service. |
| Shah et al[35] | 1998 | Longitudinal | 11 | Gujarati people over 65 years living in Leicester, UK | Community | Seven of the 11 followed up (64%). Diagnosis of dementia was reconfirmed in 6 out of 7 cases (86%) and there was evidence of further cognitive decline. |
| Shah[36] | 1999 | Case study—descriptive methodology | 12 | Gujarati patients (aged 65–90 years) seen by Gujarati psychogeriatrician | Psychogeriatric unit | n=4 (33%) with diagnosis of dementia. Difficulties interviewing Gujarati patients reported. Identifying cognitive signs and symptoms reported as most difficult. Few patients could speak English and majority could not read or write. |
| Uppal et al[37] | 2014 | Qualitative | 28 | Sikh participants aged over 18 years. Able to speak either Punjabi and/or English | Community | Three key themes: awareness and interpretation of the characteristics of dementia; multiple perspectives of the same symptoms and causes of dementia. |
| Turner et al[38] | 2005 | Qualitative | 192 | n=96 (50%) South Asian, aged 58–85 years | Community | South Asian people had less specific knowledge of dementia and believed that dementia was a normal part of ageing. Also less likely to think that medical treatment was available. Care was seen as provided by the family in the first instance. |
| Redelinghuys and Shah[39] | 1997 | Cross-sectional study | 235 | n=39 (17%) South Asians, aged 65–95 years using a geriatric psychiatry service in South London | Geriatric psychiatry | n=6 (15%) of the South Asian group had dementia compared with n=43 (22%) of the white British elders. There were no differences found between the two groups in terms of use of health and social services. |

AMT, Abbreviated Mental Test; BEHAVE-AD, Behavioural Pathology in Alzheimer's Disease Rating Scale; BEMI-C, Barts Explanatory Model Inventory Checklist; BEMI-D, BEMI-Dementia; GMS-A, Geriatric Mental State; GP, general practitioner; MMSE, Mini-Mental State Examination; NA, not available.

Asian participants.[41] The authors do not use the term South Asian but rather Asian. However, we have included this paper as it is clear from the categorisation of other ethnic groups included in the study that Asian refers to participants from South Asia who make up 2% of the total study sample. The authors interviewed 12 Asian participants using the Geriatric Mental State Examination and found a prevalence of dementia of 9% among the English-speaking sample.

Two articles reported characteristics of ethnic elders from the Indian subcontinent living in west London and attending a psychogeriatric service.[31 39] Both published reports were from the same sample population, the first consisting of cross-sectional data from a census conducted in August 1995,[39] and the second a 2-year study of all new cases entering the service between August 1995 and July 1997.[31] Sample sizes were small with South Asians making up only 12%[31] and 17%[39] of the total number of recruited participants. Dementia was diagnosed by physicians and eligible patients were identified by case note review. Redelinghuys and Shah[39] found that only a small subsample of participants had dementia, with no significant difference found between those from the Indian subcontinent (n=6, 15%) and white British elders (n=43, 22%). However, Odutoye and Shah[31] reported that over that time period there was a significant difference in the incidence of dementia with elders from the Indian subcontinent being less likely to have a dementia.

Haider and Shah conducted a study of the behavioural and psychological signs and symptoms of dementia using the Behavioural Pathology in Alzheimer's Disease Rating Scale.[21 44] They compared a sample of physician-diagnosed patients from the Indian subcontinent with a white British sample living in West London who had been referred to a day hospital for treatment. They did not find any major differences between the two groups.

In summary, studies have attempted to identify the prevalence of dementia in South Asians living in the UK. Most have reported that there is no difference in the prevalence between South Asian and white British participants. However, sample sizes in these studies have been unanimously small and the studies were conducted over a period of 14 years, therefore it is difficult to draw conclusions about the current definitive prevalence. Several studies do not report sufficient detail on the methods and instruments by which dementia was diagnosed and if there was any translation or cultural adaptation.[21 31 39] Bhatnagar and Frank noted that there is a lack of cultural adaptation of dementia measures for South Asians, which may result in high false-positive and false-negative diagnoses, which may result in unreliable prevalence rates.[18]

### Diagnosis, validation and screening
Three papers were identified that discussed either the diagnosis of dementia in the UK South Asians or validation of screening tools. Two studies were identified that

had looked to validate the Mini-Mental State Examination (MMSE) dementia screening instrument for use in the UK South Asian population. Lindesay et al investigated whether a Gujarati version of the MMSE could be used as a screening instrument in a Gujarati population in Leicester.[28] The authors reported that they followed the precedent of Ganguli et al in developing this Gujarati version of the MMSE.[45] They initially selected the items using consensus, translation and back translation, pretesting and validation against psychiatric diagnosis.[28] Lindesay et al reported that the MMSE performed well in the identification of moderate-to-severe dementia but was less effective in detecting cases of mild dementia in the Gujarati group when compared with a group of white British participants. In a second study, using the same sample as Lindesay et al,[28] Shah et al investigated the stability of dementia diagnosis over time.[35] They reported that the diagnosis of dementia was stable at follow-up over a period of >2 years (26–32 months), which indicates that the measure Gujarati MMSE has test–retest reliability in this population.

Rait et al tested the MMSE in a population of South Asians, including Gujarati and Pakistani participants.[33] The authors state that, as there were no definitive guidelines available for translating questionnaires such as the MMSE, they engaged both academics and members of the South Asian community to assess the cultural sensitivity of the items in three groups. The items were translated by a professional translation group and then back-translated with the South Asian community group. They reported that modifications were made to the MMSE, which centred around education and literacy and the modified MMSE was tested in a pilot study.[46] They found high levels of sensitivity and specificity for the MMSE, but with a lower cut-off score for the identification of mild dementia in the South Asian group.

Findings indicate that the Gujarati version of the MMSE may be effective at identifying dementia, especially where the diagnosis is certain and therefore symptoms are moderate to severe. However, it may be less effective in identifying dementia where symptoms are mild or diagnosis is uncertain.

### Knowledge, understanding and attitudes
We identified six studies that had set out to identify knowledge, understanding and attitudes to dementia in the UK South Asian community. Knowledge about dementia in Indian (n=91) and white UK/Irish/European (n=55) people was assessed using the Dementia Knowledge Questionnaire (DKQ) by Purandare et al.[32] They found that both groups had poor knowledge of causes and symptoms, and Indian older people had significantly less 'basic knowledge' about dementia. Here, the authors define 'basic knowledge' as including knowledge of epidemiology, aetiology and symptomatology as measured by the DKQ.[47] A further series of focus groups investigated general population knowledge of dementia with 28 Sikh participants.[37] Again they

found poor knowledge of dementia and that Sikh participants placed greater emphasis on physical illnesses than disorders such as dementia. La Fontaine *et al* held focus groups with a population sample of 49 Punjabi Indian participants.[25] One of their key findings was the failure of participants to directly refer to dementia or even to use lay terms to discuss it, which the authors conclude indicates a lack of recognition of dementia as a concept.

Turner *et al* explored knowledge and attitudes to dementia in a qualitative community study of South Asian (n=96) and white older people (n=96) and also found reduced knowledge of dementia in the South Asian group. Furthermore, their findings highlighted differences in attitudes to dementia care, with the South Asian group reporting that care should be provided by family and friends.[38] In interviews with 10 South Asian family carers, Lawrence *et al* looked at traditional versus non-traditional caregiver ideology.[26] The majority of South Asian carers possessed a traditional ideology in that they saw care giving as natural, expected and virtuous. However, where understanding of dementia is poor problems can occur within the family caring relationship due to beliefs about the causes of dementia in the South Asian community and the attribution of blame on to the person with dementia.[16]

Adamson interviewed 15 South Asian carers living in five cities across the UK.[16] The study looked at the relationship between individual experiences, cultural factors and social structures within this minority population. Carers tended to interpret their caring roles as an expected part of their cultural heritage and a continuation of their family relationships. They also likened the experiences of this group to other chronic illness care and suggest that the understanding of the experiences of chronic care may be useful in understanding experiences in informal care for dementia.

In summary, several studies have explored the knowledge of dementia, and beliefs and attitudes to care in the UK South Asians. All report that knowledge is poor and this group is often more focused on physical illness rather than conditions such as dementia. This finding that this group does not identify with dementia as a concept further highlights the need for valid and reliable measures to be developed to identify and then diagnose dementia in this group.

Our findings show that care is preferred to be provided by family and within the community and therefore, the UK South Asians may not access formal healthcare for dementia.

Giebel *et al* have adapted the Barts Explanatory Model Inventory Checklist (BEMI-C) for use with South Asian ethnic minority groups (BEMI-Dementia (BEMI-D)).[42] They conducted 25 qualitative interviews with South Asians and identified 123 new perceptions around their understanding of dementia. These were added to the BEMI-C to create a new checklist (BEMI-D) to better assess the barriers to dementia service uptake for this group in the future.

## Help-seeking

Five papers were identified that focused on different aspects of help-seeking for dementia in the UK South Asian population.

Two studies had looked at barriers to help-seeking for dementia. First, in a community sample of English-speaking or Bengali-speaking UK South Asians who did not have a diagnosis of dementia, Mukadam *et al* identified the barriers and facilitators to help-seeking for memory problems.[30] They identified four main categories of barriers, which interacted to prevent timely diagnosis of dementia: barriers to help-seeking for memory problems; the threshold for seeking help for memory problems; ways to overcome barriers to help-seeking; what features an educational resource should have. Second, Hailstone *et al* devised and validated a theory of planned behaviour (Attitudes of People from Ethnic Minorities for Help-seeking for Dementia) questionnaire to predict medical help-seeking for dementia in the UK South Asians.[22] They found that attitudes to dementia predicted 77% of variance in help-seeking with the strongest predictor being perceived social pressure.

Furthermore, there were two studies which had specifically identified religious explanations as barriers to help-seeking for dementia in South Asians. Mackenzie interviewed 11 Pakistani and 5 Indian carers and found that stigma resulted from religious and magical beliefs around the causes of dementia and resulted in concealment from their own community and delays in help-seeking.[29] Giebel *et al* looked at the differences in the perceptions of South Asians who do and do not consult a general practitioner (GP) about dementia.[20] They found those who did not consult a GP were significantly more likely to consider memory problems as given by God, with the view that medical intervention was therefore inappropriate. In a case study of a Muslim, Pakistani patient accessing healthcare for dementia in the UK, Regan further highlighted the importance of understanding a person's religious community and its role in providing both support and reducing stigma and isolation.[40]

The included studies highlight that attitudes and beliefs about dementia can serve as barriers to accessing healthcare, in particular understanding dementia within a religious context can delay help-seeking. However, religious communities can also play a vital role in supporting patients with dementia and in reducing stigma. This may be the case across other ethnic minority communities in the UK, not just South Asians. It is therefore important to learn more about the religious context in which people understand dementia and for healthcare providers to engage with local communities and religious leaders in order to work in partnership with them to increase support and reduce stigma.

## Experiences of dementia

We identified three articles that looked at understanding the experience of dementia for the UK South Asians. Bowes and Wilkinson explored the experiences

of South Asian patients with dementia in Scotland.[19] Interviews were conducted with 11 health professionals, and 4 case studies were built up through multiple interviews with four people with dementia, as well as interviews with their family and carers. The authors reported overwhelmingly negative experiences of dementia and of health services from the case studies. From the interviews with health professionals, the authors concluded that there was a need to develop and promote culturally sensitive services. Furthermore, the article by Jutlla looked at the experiences of 12 Sikh carers, caring for a family member with dementia, in Wolverhampton.[23] In particular, they looked at the influences of migration experiences and migration identities. They found that knowledge of a person's background and experiences is important for understanding how they then experience health services and caring roles. This is presented as an advocate for culturally appropriate services and for person-centred dementia care.

Lawrence *et al* looked at experiences of dementia in three ethnic groups, 11 black Caribbean, 9 South Asian and 10 white British older people with dementia.[27] The paper reported a comparison in the personal experiences of the condition in the three groups. They reported similar themes across groups with a key finding being that patients assessed their condition by the degree to which it interfered with 'valued elements of life'. The authors concluded that development of culturally sensitive approaches to care should promote roles, relationships and activities that the patient values.

### Service organisation and delivery

We found three articles that discussed service organisation and delivery for South Asian patients with dementia in the UK. First, in 1999 Shah, a Gujarati Psychogeriatrician, published a descriptive account of his own experiences working with 12 Gujarati patients in their language.[36] Four of the patients had a diagnosis of dementia. He identified particular problems in eliciting cognitive symptoms due to problems with the translation and the lack of culturally appropriate assessments.[36] Second, Kaur *et al* wrote about the role an Asian link nurse for Punjabi-speaking people of Asian origin in a dementia service in Wolverhampton.[24] The authors reported success in providing appropriate and culturally sensitive help and information for healthcare professionals, voluntary services and South Asian patients with dementia. Third, Seabrooke and Milne discussed the Dementia Collaborative Project in North West Kent, which aimed to raise awareness of memory problems in South Asians and facilitate access to screening and diagnosis.[34] The authors reported an increase in referrals to a specialist clinic for memory assessments, some of which were for South Asian patients, and an increase in health professional's knowledge of memory problems in South Asians.

## DISCUSSION

We have conducted the first scoping review of dementia in the UK South Asian population. We identified 27 studies that were focused on one of the following areas: prevalence, diagnosis and screening, knowledge and attitudes, help-seeking, experience and service organisation and delivery for dementia in the UK South Asian population. Overall, this review highlights that research on dementia in UK South Asians is limited. Studies on prevalence are outdated, we lack culturally adapted instruments to diagnose dementia, and community engagement work is in its infancy.

There have been few studies of the prevalence of dementia within the UK South Asian population, and where studies have set out to identify prevalence, sample sizes have been small. Furthermore, prevalence studies are old, with the most recent being published in 2004 and therefore, the current prevalence of dementia in the UK South Asians in unknown.[21] There is a need for large epidemiological studies of dementia in the UK South Asian population in order to confirm the prevalence nationally. There is also a need for studies embedded within epidemiological work that can explore the current experience, both quantitatively and qualitatively, of UK South Asians with dementia in the context of dementia healthcare. Future cohort studies of this nature should provide data grouped by ethnicity and ensure that ethnic groups are well defined to allow results to be included in future reviews.

We found only two studies that aimed to validate the MMSE screening tool within the UK South Asian population.[28 33] These two studies limited South Asians to a screening process with no access to receiving a confirmed diagnosis of dementia. We did not identify any studies that had addressed the introduction and validation of a purely diagnostic assessment for the UK South Asians, such as the Addenbrooke's Cognitive Examination Version III or the Motreal Cognitive Assessment.[48 49] This compromises current knowledge of dementia prevalence within UK South Asians, as without a culturally appropriate, validated diagnostic assessment individuals cannot be diagnosed. The lack of a culturally adapted and validated diagnostic tool risks that the UK South Asians who enter into the diagnostic process will receive higher rates of false-positive or false-negative scores. This invalidates the current diagnoses given to UK South Asian patients. We suggest that undertaking the cultural adaptation and validation of diagnostic assessments according to the published guidelines would improve the diagnostic process.[50]

Most of the existing research in this area has been conducted in the community, which reflects the fact that UK South Asians do not regularly access formal healthcare services for memory problems. We need to develop culturally sensitive screening tools, diagnostic tests and interventions, whilst also engaging communities. It is important to develop interventions in parallel with community engagement to ensure that there are culturally appropriate services ready to be accessed if engagement is successfully increased.[7]

We were unable to identify a single clinical trial of an intervention for dementia in the UK South Asian population, either among patients or carers. This highlights a critical gap in ongoing dementia research and indicates a neglect of 5.3% of the UK population who identify as South Asian. This neglect calls into question whether it is appropriate to implement existing interventions for this ethnic group without any assessment of their feasibility, acceptability and effectiveness.[3] This review has identified a number of different views and perceptions held about dementia by UK South Asians. Therefore, existing interventions need to be adapted, or new culturally sensitive interventions should be developed and trialled, to accommodate for these differences. From our findings relating to service organisation and delivery, we can see strategies emerging that may increase engagement from this population, such as the ethnic matching of staff, and increasing engagement work with South Asian community. These strategies should now be considered when designing clinical trials to test culturally adapted interventions.

This review has highlighted factors that impact on our ability to conduct intervention trials in this population; these may include issues surrounding diagnosis and the lack of appropriately validated diagnostic instruments, as well as attitudes and beliefs about dementia, which may affect recruitment. Much of the qualitative work identified in this review has highlighted the religious explanations by which the South Asian community often understand symptoms of dementia. Studies report that this causes a barrier to help-seeking and will result in delayed treatment. Furthermore, due to religious explanatory models for patients with dementia family carers often perceive engagement with treatment as inappropriate. Culturally sensitive community engagement work is needed to engage the South Asian community and encourage understanding of dementia. New interventions should also acknowledge and include the family approach to care, which is seen as of paramount importance in the South Asian community.

This review is the first review to scope all of the literature on dementia in the UK South Asian population. We scoped the literature in line with the published guidelines for scoping reviews and conducted a systematic search of the literature.[13 15] However, it is possible that there are other studies that were not identified as part of this search; in part this could be due to a lack of standardisation of the terminologies used in the literature. One potential limitation of our scoping review is that we restricted our eligible studies to those conducted in the UK. Although narrow, our focus on the UK South Asians allowed us to identify gaps in the literature for a group that may face specific challenges and barriers to accessing healthcare in the National Health Service. It would, however, be beneficial to review studies of dementia in South Asians globally in order to identify key learning for development of any intervention.

The scoping review is not designed to provide a definitive answer to questions in any of the key themes discussed in this review because there are important differences across the included studies, which were not extensively evaluated (eg, quality, design and population). While this is consistent with published guidelines for scoping reviews, it means that we are unable to identify any gaps in the literature that arise due to poor quality research.[13 14] The purpose of this review was to map the literature and to provide a useful framework to guide future research directions. The scoping review methodology is particularly suited to this aim as it provides a comprehensive account of current progress and challenges, which can be used to develop future research and priorities to improve care for the UK South Asians.[13]

## CONCLUSION

A number of studies have been published on dementia in the UK South Asian population. However, studies of prevalence, diagnosis, and service organisation and delivery are limited. We found no clinical trials of culturally appropriate interventions for South Asians with dementia in the UK. The existing evidence comes from small-scale service evaluations and case studies.

Future research efforts should concentrate on developing and validating culturally appropriate diagnostic tools for the UK South Asians. Epidemiological studies are needed to accurately identify the prevalence of dementia in this group. Cultural adaptation and clinical trials of appropriate culturally sensitive interventions are needed to run in parallel to diagnosis and community engagement work to ensure there are effectiveness and acceptable treatments for South Asians to access once identified with dementia.

**Contributors** AB and CK ran the search and screened papers. AB, CK, GD-W, MP, WW identified and agreed eligible papers. AB wrote the paper and CK, GD-W, MP, NM, WW edited and revised the paper for critical content.

**Funding** This paper represented independent research funded by the National Institute for Health Research (NIHR) School for Primary Care Research (SPCR), grant number SPCR 300.

**Disclaimer** The views expressed are those of the author(s) and not necessarily those of the National Health Service, the National Institute for Health Research or the Department of Health.

**Competing interests** None declared.

**Patient consent** Not required.

**Provenance and peer review** Not commissioned; externally peer reviewed.

**Data sharing statement** Data is available from the corresponding author on request.

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
