## [Reviewer comments · BMJ Open]

ARTICLE DETAILS

TITLE (PROVISIONAL)	Dementia in UK South Asians: A scoping review of the literature
AUTHORS	Blakemore, Amy; Kenning, Cassandra; Mirza, Nadine; Daker-White, Gavin; Panagioti, Maria; Waheed, Waquas

VERSION 1 – REVIEW

REVIEWER	Naaheed Mukadam UCL, England
REVIEW RETURNED	31-Oct-2017

GENERAL COMMENTS	This is an interesting and under-researched field and the authors have done an admirable job of sifting through the literature to find relevant papers. The division of research paper categories into those addressing prevalence, screening, experience etc is a useful one and helps to condense a vast body of literature into easily understood chunks. I have a few minor suggestions. Firstly, the paper has omitted two prevalence studies which included South Asian people with dementia (McCracken et al 1997 and Livingston et al 1990). Again, these are small samples and dated but they are relevant to the literature. It is concerning that such a broad search strategy missed these papers, particularly the McCracken one. Perhaps the authors could comment on this and include these papers if they deem them suitable. Currently the paper is sub-divided into types of research studies, e.g. prevalence, diagnosis etc. Each section has a short summary at the end of what has been found and this is useful. It might help to have another section after this highlighting what is still not known in this field, which the authors can tie together in their final discussion.
---

REVIEWER	Anne-Le Morville Dept. of rehabilitation Jönköping University Sweden
REVIEW RETURNED	20-Nov-2017

GENERAL COMMENTS	BMJ Open Review Dementia in UK South Asians: A scoping review of the literature
---

Dear authors

Thank for your well-written and interesting manuscript. I find this an interesting subject and worthwhile to research. I must admit that I find your manuscript lacking more focus, so therefore I have suggested areas where an elaboration would enhance your manuscript.

I know that you have a restriction on length of the manuscript, but as these are usually negotiable when writing a review, I suggest that you discuss the length of your manuscript with the editors, as it needs elaborations on some topics.

Abstract

Your abstract does not give any information on your findings except for six categories.

Introduction

However, I have a hard time to figure out why this is a problem, when I read your introduction. I can make more or less qualified guesses, but you need further elaboration on the problems of dementia in your target group. The lack of information in this part makes it hard to see what you're looking for, which is also reflected in your aim.

Is there, or will there be a larger economic burden regarding your target group?

How do culture/religious beliefs influence the treatment and service provision?

How do you define older people?

You write "potential difficulties", but what are these difficulties except for language, and what is known?

You also write that you want to identify gaps in the literature, but this should be more specific. I also think that when you are questioning the (lack of) literature, you should refer to research regarding the problems with using instruments developed for middleclass persons

in Western countries.

Regarding references in this part of your manuscript you refer to statistics (3) from 2012. As the world changes rapidly within 5 years, I would suggest that you search for new material.

Methods

Your choice of method is appropriate, but I lack clarity in this part. Your description of the scoping review process is very brief, and I would like if you could argue why you choose this method. I do find Arksey & O'Malley and Levac appropriate, but there are new and more specified literature regarding this method, which may help you describe the method more thoroughly.

You contradict yourself as you write that you did not exclude literature based on method and two lines down you exclude systematic reviews and narrative literature reviews, so please decide what you did and describe in more detail why.

I also find your lack of limitation regarding year of publication when looking at prevalence problematic. However, this also depends on your focus, as is it the prevalence or lack of instruments to measure prevalence.

I have problems following your screening process. Did you exclude based on titles alone, or abstracts and where in the process did you do this? So please expand your flow-chart or write in the text how you did this.

Were the themes you represent identified beforehand?

Results

Prevalence and characteristics

The first study (12) mentioned the problems of using the scale for testing for dementia, but you do not refer to methodological issues in the next (25,33). Which instruments did they use to diagnose dementia?

	Did Haider et al discuss the data-gathering method? In your summary you do not touch upon the methodological issues, which is an important part when discussing prevalence. Diagnosis, validation and screening The results around the MMSE are interesting, but I would like to know a bit more about the validation, as there are several pit-falls and ways of validating questionnaires that may give an incorrect answer. Have the tests e.g. been tested for face and content validity, reliability and responsiveness, which translation method did they use etc.? Knowledge, understanding and attitudes The finding that the target group do not recognize dementia as a concept further enhances my point of not having valid instruments as a severe problem. Service organisations and delivery I find that the results regarding the positive outcome of raising awareness and use of culturally “safe” health care personnel are very interesting and also the lack of knowledge in the health professionals knowledge/understanding. Discussion/Conclusion I think that you need to discuss your method in greater detail. Both regarding language, both also the fact that you limited yourself to UK studies, which makes sense if you only look at prevalence, but as you look at a broader perspective, this is a shortcoming. I find your discussion interesting and fulfilling, and think that a few lines on ideas of which methods to use in the future research, developing both questionnaires and appropriate intervention methods, would be interesting.
--	--

VERSION 1 – AUTHOR RESPONSE

Reviewer 1 Comments	Author Responses
-------------------------

1	Firstly, the paper has omitted two prevalence studies which included South Asian people with dementia (McCracken et al 1997 and Livingstone et al 1990). Again, these are small samples and dated but they are relevant to the literature. It is concerning that such a broad search strategy missed these papers, particularly the McCracken one. Perhaps the authors could comment on this and include these papers if they deem them suitable.	We thank the reviewer for identifying these papers (McCracken et al. 1997 and Livingstone et al. 1990) We have reviewed our search and identified that McCracken et al. (1997) was returned in our initial search but was screened out as it did not have a specific focus on South Asian participants. We have now reviewed the paper again and confirm that this was an omission. We have identified that McCracken et al. (1997) does in fact include prevalence data for Asian participants. Therefore, we have added the paper to our review and updated the text in the results section, results table and recruitment figures and flowchart. There are two papers by Livingstone et al. (1990) which discuss the Gospel Oak Study. Having reviewed our search results these were indeed not returned by our search strategy. We have now reviewed both of these papers and have reached the conclusion that they are not eligible for inclusion in our review as they do not specifically report data on dementia in the UK South Asian population. We have updated our recruitment flowchart and figures quoted in the text to reflect that we have reviewed these two additional papers. In addition to this we have included a discussion point that with authors of future studies should better define ethnic groups and provide group based data for analysis (Lines 474-476).
2	Currently the paper is sub-divided into types of research studies, e.g. prevalence, diagnosis etc. Each section has a short summary at the end of what has been found and this is useful. It might help to have another section after this highlighting what is still not known in this field, which the authors can tie together in their final discussion.	We thank the reviewer for their comments. We have reviewed our results section and discussion and conclude that the addition of a section explaining what is still not known in this field within the results section would be a repeat of the summary of results and direction for future research which is already included in the discussion. We feel that this is best placed in the discussion and therefore, we have not included this additional section. However, if the reviewer and editor feel that the discussion is not sufficient and an additional section is necessary and are willing to allow us to increase the word count further then we would be happy to reconsider.

	Reviewer 2 Comments	Author Responses
1	Thank for your well-written and interesting manuscript. I find this an interesting subject and worthwhile to research. I must admit that I find your manuscript lacking more focus, so therefore I have suggested areas where an elaboration would enhance your manuscript. I know that you have a restriction on length of the manuscript, but as these are usually negotiable when writing a review, I suggest that you discuss the length of your manuscript with the editors, as it needs elaborations on some topics.	Thank you for your positive feedback on our manuscript. We take your point about clarifying the focus of this review and have addressed this in our responses to your comments (below) and changes to the manuscript.
2	Your abstract does not give any information on your findings except for six categories.	We have reviewed the abstract (lines 54-66) and altered the results section to clarify that the findings from our scoping review are that studies of dementia in UK south Asians are very limited to date. The aim of our review is to identify the gaps in the literature and make a case for future research that is needed in the future which we move on to in the conclusion section. We hope this satisfies the reviewers comment. However, we would be happy to consider further changes should this be necessary.
3	However, I have a hard time to figure out why this is a problem, when I read your introduction. I can make more or less qualified guesses, but you need further elaboration on the problems of dementia in your target group. The lack of information in this part makes it hard to see what you're looking for, which is also reflected in your aim. Is there, or will there be a larger economic burden regarding your target group? How do culture/religious beliefs influence the treatment and service provision?	Thank you for this useful comment. We have now considerably amended the introduction to improve the clarity of our argument around why this scoping review is so important at this time (please see lines 117-142). We have also clarified our aim and defined the purpose of scoping review (Lines 144-153). We have improved the paragraph which includes our aim to make it clear what it is we hope to gain from conducting this scoping review.

	How do you define older people? You write “potential difficulties”, but what are these difficulties except for language, and what is known?	We have defined older people as those over the age of 65. We have added clarification of this within the introduction and added the appropriate reference (Line 126). We have removed the phrase ‘potential difficulties’ and added clarification about where we see the key difficulties. The sentences that follow outline these difficulties in more detail (lines 122-125).
4	You also write that you want to identify gaps in the literature, but this should be more specific. I also think that when you are questioning the (lack of) literature, you should refer to research regarding the problems with using instruments developed for middleclass persons in Western countries.	We have significantly rewritten the final paragraph of our introduction to specify what we mean by gaps in the literature and to speculate about how this information can be used to inform future research with appropriate references (lines 142-151). We have added a sentence within the third paragraph of our introduction which highlights the lack of appropriately translated and culturally adapted screening and diagnostic tools for dementia in the South Asian population (123-127).
5	Regarding references in this part of your manuscript you refer to statistics (3) from 2012. As the world changes rapidly within 5 years, I would suggest that you search for new material.	We have reviewed reference 3 as suggested and can confirm that this is the latest release from the Office for National Statistics on Ethnicity and National Identity in England and Wales https://www.ons.gov.uk/peoplepopulationandcommunity/culturalidentity/ethnicity/articles/ethnicityandnationalidentityinenglandandwales/2012-12-11).
6	Your choice of method is appropriate, but I lack clarity in this part. Your description of the scoping review process is very brief, and I would like if you could argue why you choose this method. I do find Arksey & O’Malley and Levac appropriate, but there are new and more specified literature regarding this method, which may help you describe the method more thoroughly.	In the first paragraph of the methods section we have referred to more recent literature by Pham et al. (2014) on the use of the Arksey and O’Malley framework (lines 167-174).
7	You contradict yourself as you write	Thank you for noticing this error (lines 179-184). We have

	that you did not exclude literature based on method and two lines down you exclude systematic reviews and narrative literature reviews, so please decide what you did and describe in more detail why.	now amended this section to read: 'We did not exclude studies based year of publication. Published dissertations were included but any unpublished dissertation was not. Systematic reviews and narrative literature reviews were not included but we hand search the reference lists of all relevant reviews that were returned in the search in order to identity additional primary studies for inclusion. Conference proceedings were not included.'
8	I also find your lack of limitation regarding year of publication when looking at prevalence problematic. However, this also depends on your focus, as is it the prevalence or lack of instruments to measure prevalence.	Thank you for this comment. There are existing sentences both in prevalence section of the results and the second paragraph of the discussion which highlight that the data on prevalence are old. The most recent study was published in 2004. In order to improve the clarity of this point we have added a further sentence to the discussion to clarify that the current prevalence of dementia in UK South Asians is unknown (lines 466-470). We hope this has clearly resolves this point but would be happy to make further amendments if necessary.
9	I have problems following your screening process. Did you exclude based on titles alone, or abstracts and where in the process did you do this? So please expand your flow-chart or write in the text how you did this.	We initially screened based on titles and abstracts. In the methods section we state (lines 210-212): 'Titles and abstracts of all citations were first screened by author (AB), those that were not related to dementia and South Asians, or had been conducted outside of the UK, were discarded.' We have amended the flow chart in Figure 1 to clearly state this.
10	Were the themes you represent identified beforehand?	No, the themes were identified once we had the final pool of eligible papers based on the literature that was available. We have amended the methods section to clearly state this (lines 226-229).
11	Prevalence and characteristics The first study (12) mentioned the problems of using the scale for testing for dementia, but you do not refer to methodological issues in the next (25,33). Which instruments did they use to diagnose dementia? Did Haider et al discuss the data-	Odutoye et al. (1999) and Redelinghuys et al. (1997) are studies of patients referred to a psychogeriatric service and data on diagnoses were extracted from case notes. Dementia was diagnosed by physicians prior to the studies. The Haider et al. (2004) is a study of patients admitted to a day hospital for assessment and treatment of dementia after assessment at home by a physician. The assessment measure is not discussed.

gathering method? In your summary you do not touch upon the methodological issues, which is an important part when discussing prevalence.	We have amended the paragraph on each of these three studies and have added a comment on diagnosis to the summary paragraph (lines 248-298). Thank you for noticing this. We have added this comment into the summary paragraph on prevalence (lines 295-298)
1 2 . Diagnosis, validation and screening The results around the MMSE are interesting, but I would like to know a bit more about the validation, as there are several pit-falls and ways of validating questionnaires that may give an incorrect answer. Have the tests e.g. been tested for face and content validity, reliability and responsiveness, which translation method did they use etc.?	We have found this comment very useful and as a result have significantly reworked our results section on Diagnosis, validation and screening. We have added additional information about the validation of the questionnaires. Please see lines (300-329).
1 3 . Knowledge, understanding and attitudes The finding that the target group do not recognize dementia as a concept further enhances my point of not having valid instruments as a severe problem.	Thank you for this comment. We have added a sentence to highlight this point in this section (lines 369-372).
1 4 . Service organisations and delivery I find that the results regarding the positive outcome of raising awareness and use of culturally “safe” health care personnel are very interesting and also the lack of knowledge in the health professionals knowledge/understanding.	Thank you for your comment.
1 5 . Discussion/Conclusion I think that you need to discuss your method in greater detail. Both regarding language, both also the fact that you limited yourself to UK	Thank you, we have added a comment to the final paragraph of the discussion to consider the issue of restricting our review to UK studies only.

	studies, which makes sense if you only look at prevalence, but as you look at a broader perspective, this is a shortcoming.	
1 6 .	I find your discussion interesting and fulfilling, and think that a few lines on ideas of which methods to use in the future research, developing both questionnaires and appropriate intervention methods, would be interesting.	We are pleased that you find the discussion both interesting and fulfilling. Throughout our discussion we have made recommendations for future research. We have discussed the need to develop large epidemiological studies of dementia in UK South Asians to assess prevalence (lines 477-483). We have suggested that these studies be coupled with quantitative and qualitative evaluation of dementia health care use in this population (lines 478-482). We have added the recommendation that screening and diagnostic tools need to be culturally adapted and validated according to guidelines for cultural adaptation with appropriate reference to a recently published work in this area. We have added further discussion on how to engage the South Asian community (lines 515-519). We suggest that we need to culturally adapt existing interventions or develop new culturally sensitive interventions (lines 513-514). We highlight that emerging engagement strategies such as, ethnic matching of staff, should be considered when designing and conducting clinical trials in this area (lines 515-519). We move on to highlight potential methodological problems when conducting trials in this group (lines 521-532). Finally, we suggest that interventions need to be developed in parallel with community engagement work to ensure that there are culturally appropriate services ready to be accessed if engagement is successfully increased (576-579).

VERSION 2 – REVIEW

REVIEWER	Naaheed Mukadam UCL, UK
REVIEW RETURNED	26-Jan-2018

GENERAL COMMENTS	Thank you for addressing concerns raised by reviewers.
--

REVIEWER	Anne-Le Morville Departement of Rehabilitation Jönköping University Sweden
REVIEW RETURNED	31-Jan-2018

GENERAL COMMENTS	Dear Authors You have made a thorough revision and I can only recommend your manuscript for publication. Your manuscript is more focused and I like that you have made very clear and precise conclusions. Happy publishing!
---